# Pharmaceutical Payments to Japanese Board-Certified Infectious Disease Specialists: A Four-Year Retrospective Analysis of Payments from 92 Pharmaceutical Companies between 2016 and 2019

**DOI:** 10.3390/ijerph19127417

**Published:** 2022-06-16

**Authors:** Anju Murayama, Sae Kamamoto, Hiroaki Saito, Kohki Yamada, Divya Bhandari, Iori Shoji, Hanano Mamada, Moe Kawashima, Erika Yamashita, Eiji Kusumi, Toyoaki Sawano, Binaya Sapkota, Tetsuya Tanimoto, Akihiko Ozaki

**Affiliations:** 1Medical Governance Research Institute, Minato-ku 108-0074, Tokyo, Japan; sae.joy1056210@gmail.com (S.K.); kykohki7691@gmail.com (K.Y.); rayordeal3@gmail.com (D.B.); istouchgo@gmail.com (I.S.); hananomamada.a.1005@gmail.com (H.M.); moek199903@gmail.com (M.K.); e.yamashita1101@gmail.com (E.Y.); tetanimot@yahoo.co.jp (T.T.); ozakiakihiko@gmail.com (A.O.); 2School of Medicine, Tohoku University, Sendai City 980-8577, Miyagi, Japan; 3School of Medicine, Hamamatsu University, Hamamatsu City 431-2102, Shizuoka, Japan; 4Department of Gastroenterology, Sendai Kosei Hospital, Sendai City 980-0873, Miyagi, Japan; h.saito0515@gmail.com; 5Navitas Clinic Shinjuku, Shinjuku-ku 160-0022, Tokyo, Japan; eiji.kusumi@gmail.com; 6Department of Surgery, Jyoban Hospital of Tokiwa Foundation, Iwaki City 972-8322, Fukushima, Japan; toyoakisawano@gmail.com; 7Nobel College Faculty of Health Sciences, Pokhara University, Kathmandu 33700, Nepal; sapkota.binaya@gmail.com; 8Department of Internal Medicine, Navitas Clinic, Tachikawa City 160-0022, Tokyo, Japan; 9Department of Breast and Thyroid Surgery, Jyoban Hospital of Tokiwa Foundation, Iwaki City 972-8322, Fukushima, Japan

**Keywords:** board-certified infectious disease specialists, pharmaceutical payments, physician payment, conflicts of interest, Japan

## Abstract

Backgrounds: Conflict of interest with pharmaceutical companies is one of the most concerned issues in infectious diseases. However, there is a lack of whole picture of detailed payments in Japan. Methods: This retrospective study assessed financial relationships between pharmaceutical companies and all infectious disease specialists board-certified by the Japanese Association for Infectious Disease, using publicly disclosed payment data from 92 major pharmaceutical companies. Descriptive analyses were conducted for the payments. Payment trends were examined by the generalized estimating equations. Results: Of 1614 board-certified infection disease specialists, 1055 (65.4%) received a total of $17,784,070 payments, corresponding to 21,680 contracts between 2016 and 2019. The mean ± SD and median (interquartile range: IQR) were $16,857 ± $45,010 and $3183 ($938–$11,250) in payments. All board executive members of Japanese Association of Infectious Disease received higher payments averaging $163,792. There were no significant changes in payments per specialist (annual change rate: −1.4% [95% CI: −4.7–2.3%], *p* = 0.48) and prevalence of specialists with payments (annual change rate: −1.4% [95% CI: −3.1–0.2%], *p* = 0.093) over the four years. Conclusion: There were substantial financial relationships between pharmaceutical companies and board-certified infectious disease specialists in Japan. Furthermore, high ranked specialists such as those in the executive board had stronger financial ties with the companies.

## 1. Introduction

There has been an increasing concern on financial relationships between healthcare professionals, healthcare organizations and pharmaceutical companies, which sometimes become conflicts of interest (COI), because of its potential bias on healthcare. There is concern that pharmaceutical companies are attempting to influence prescribing and practice activities through lobbying activities against physicians and academic activities [1]. In response to this concern, many countries have started requesting pharmaceutical companies to disclose data on their donations and honoraria to healthcare professionals and healthcare organizations [2,3]. Consequently, previous studies utilizing these data have demonstrated that there were substantial financial relationships between pharmaceutical companies and healthcare professionals [4,5,6,7]. While there have been efforts to uncover inappropriate prescribing and how to deal with it, efforts to properly manage financial conflicts of interest are ultimately the most important in the series of activities to date that encourage appropriate prescribing activities [8,9].

Proper management of financial conflicts of interest is one of the greatest concerns among infectious diseases physicians. As in the case of Lyme disease guideline issued by the Infectious Diseases Society of America (IDSA), the guideline chair and authors manipulated the guideline recommendations and statements for the benefits of testing and insurance companies, leading to inadmissible harms on the patients in the US [10].

In cases of Japan, 91.7% of authors of clinical practice guideline for methicillin-resistant *Staphylococcus aureus* issued by the Japanese Association for Infectious Diseases (JAID) received an average of $28,371 personal payments in 2016, and four pharmaceutical companies’ employees were involved in the guideline development [11]. Moreover, during coronavirus disease 2019 (COVID-19) pandemic, substantial financial relationships were uncovered among healthcare professionals specialized in infectious diseases and pharmaceutical companies worldwide. However, the Japanese government COVID-19 advisory board members did not manage to disclose the detailed information on COI with pharmaceutical companies [12]. Although COI among influential infectious disease experts such as television commentators specialized in infectious diseases [13], guideline authors [7,11], and government advisory members [12] were investigated, the whole picture of financial relationships with pharmaceutical companies remains to be elucidated. Since board-certified infectious disease specialists directly prescribe drugs for patients, it is crucial to understand financial relationships among pharmaceutical companies and those specialists.

This study aimed to elucidate the prevalence of board-certified infectious disease specialists receiving payment from pharmaceutical companies, the magnitude of the payments, and payments trend over last few years in Japan.

## 2. Methods

### 2.1. Study Design and Participants

This study was a retrospective analysis evaluating financial relationships among all board-certified infectious disease specialists and pharmaceutical companies in Japan. All infectious disease specialists who were board-certified by the JAID as of November 2021 were included in this study, as the JAID did not disclose name list of specialists for previous years. The JAID is the largest and most prestigious professional medical society for infectious diseases in Japan, which contributed to improve patient care by promoting research and training physicians for infectious diseases in Japan since its establishment in 1926. Also, the JAID is the only organization in Japan that trains and certifies infectious diseases specialists in the country.

As of November 2021, the JAID required physicians to complete several requirements to certificate them as infectious disease specialists, such as being a specialist certified by at least one of the 19 major Japanese medical societies [14], having completed at least six years of clinical practice training after having acquired a medical license and at least three years of specialized training in infectious diseases at an institution accredited by the JAID, and having published at least one academic article on a peer-review journal and at least two conference presentations as the first author.

### 2.2. Data Collection

Data concerning name and affiliations for all of those board-certified specialists were extracted from the official webpage of the JAID (https://www.kansensho.or.jp/modules/senmoni/index.php?content_id=29) on 10 November 2021. Also, the JAID webpage provided us names of all executive board members as of 2021. In order to take into account the time lag between approval timing and marketing timing by pharmaceutical companies, we considered that a predominant part of promotional activities for products approved in 2015 had started in 2016 and all drugs with additional or new indications for infectious diseases between 2015 and 2019 were extracted from the database of the Pharmaceuticals and Medical Devices Agency [15], the Japanese regulatory authority for drugs and medical devices.

Payment data from 2016 to 2019 to all healthcare professionals and healthcare organizations for lecturing, writing, and consulting were collected from all 92 pharmaceutical companies affiliated with Japan Pharmaceutical Manufacturers Association (JPMA) [16,17]. JPMA required that the member companies disclose only the payment for lecturing, writing, and consulting on the individual basis. However, payment categories such as meals, travel, and accommodations were not disclosed on each individual specialist. The companies have published, updated the payment data each year on their company webpages, and deleted the payment data for previous years when they updated latest data. The payment data for all companies belonging to the JPMA were collected from the 2016 data. As of May 2022, the payment data in 2019 were the latest analyzable data in Japan. Thus, we could only analyze personal payments concerning lecturing, writing, and consulting between 2016 and 2019.

Then, we stored all payment data collected from 92 companies into an excel file and structured the searchable payment database. By searching for the specialist names in the payment database, the payment data to the infectious disease specialists were extracted from the payment database. The extracted data included recipient names, recipient affiliations, monetary amount, number of payment contracts, payment category, and name of pharmaceutical company making the payment. To remove payment data of different persons with duplicate names in the database, we checked and compared the affiliations, affiliation address, and recipient specialties among the data from the JAID and the pharmaceutical companies. In cases where affiliation reported by the company differed from the one reported by the JAID, we manually googled the name of specialists and collected other data from the official institutional webpages and other sources to verify that they were the same person. The detailed process can be found in our previously published papers [6,16,17].

### 2.3. Data Analysis

Descriptive analyses were performed for payment values and number of contracts on individual specialist and pharmaceutical company levels. Average and median payments, contracts, number of companies making payments per specialist were calculated based on the only specialists receiving payment in each year, as in other studies assessing pharmaceutical payments to physicians [18,19,20,21]. To compare the payments among the specialists with and without a leading role in the JAID, the average mean and median payments were evaluated by the specialists with and without the executive board membership. The difference between two groups was assessed by the Mann-Whitney U test, as the payments data were not normally distributed.

The Gini index and the shares of the payment values held by the top 1%, 5%, 10%, and 25% of the specialists were calculated to examine distribution and concentration of payments. The Gini index ranges from 0 to1, and the greater the Gini index is, the greater the disparity in the distribution of payments on the specialist basis, as performed previously [22]. Also, payment distributions were geographically examined on prefectures and regions, as there were differences in number of the specialists and the medical institutions accredited by the JAID.

The population-averaged generalized estimating equations (GEE) were performed to evaluate the payment trends. As the payment distribution was highly skewed (Appendix A), negative binomial GEE model for the payment values per specialist, and linear GEE log-linked model with binomial distribution for the prevalence of specialists with payments were used. The year of payments was set as independent variable, and the payment values per specialist and proportion of physicians receiving payments were set as dependent variables. The average annual changes in independent variables, payment values per specialist and prevalence of specialists with one or more payments, were reported as a relative percentage. As several pharmaceutical companies among all 92 companies disaffiliated from the JPMA and newly joined the JPMA, there were 18 companies without payment data over the four years. Thus, the average and median payments for each year and the trend of payments were calculated based on payments from all 92 companies and 74 companies with payment data for the four years between 2016 and 2019, as previously described [4,5,23].

Finally, we assessed association between number of drugs with new or additional indications and (1) total payments and (2) number of specialists with payments on company level using the Spearman’s correlation.

Japanese yen (¥) was converted into US dollars ($) using 2019 average monthly exchange rates of ¥109.0 per $1. All analyses were conducted using Microsoft Excel, version 16.0 (Microsoft Corp., Redmond, WA, USA) and Stata version 15 (StataCorp, College Station, TX, USA).

## 3. Ethical Approval

The Ethics Committee of the Medical Governance Research Institute approved this study (approval number: MG2018-04-20200605; approval date: 5 June 2020). As this study was a retrospective analysis of the publicly available information, informed consent was waived by the Ethics Committee.

## 4. Results

We identified 1614 infectious disease specialists certified by the JAID as of 10 November 2021. The JAID stated that there were a total of 1622 infectious disease specialists in Japan, and therefore, names of eight specialists missing were not disclosed on the webpage, as the specialists could have wished not to disclose their names on the webpage.

### 4.1. Overview and Per-Specialist Payments

Of 1614 eligible board-certified infectious disease specialists, 1055 (65.4%) received one or more payments, totaling $17,784,070 corresponding to 21,680 payment counts between 2016 and 2019 (Table 1). Among 92 companies, 78 (84.8%) made at least one payment to the specialists over the four-year period. The four-year average (standard deviation: SD) and median (interquartile range: IQR) were $16,857 ($45,010) and $3183 ($938–$11,250) in payments; 20.5 (41.6) and 6.0 (2.0–19.0) in payment contracts; and 5.6 (5.2) and 4.0 (2.0–8.0) in number of pharmaceutical companies per specialist (Table 1).

Regarding the payment distribution, although 34.6% of specialists had no payments, 5.1% and 2.7% received more than $50,000 and $100,000, respectively. The Gini index for the four-year cumulative payments per specialist was 0.86. Top 1%, 5%, 10% and 25% of the specialists occupied 26.3% (95% confidence interval (CI): 21.4–31.2%), 61.5% (95% CI: 57.0–65.9%), 77.2% (95% CI: 73.9–80.4%), and 93.6% (95% CI: 92.5–94.7%) of total payments, respectively (Appendix A). One specialist received a maximum of $711,965 payments over the four-year from 21 pharmaceutical companies.

Of 18 executive members of the JAID as of November 2021, 17 (94.4%) had certification of infectious disease specialists. All the 17 members with the specialist certification, including the current JAID president, received more substantial payments averaging $163,792 (SD: $173,475; median: $95,551; IQR: $54,227–$207,948) than the specialists without executive board membership (*p* < 0.001 in Mann-Whitney U test) over the four years.

### 4.2. Payment Trend between 2016 and 2019

The average annual payments per specialist ranged from $5775 (SD: $13,410) in 2017 to $6134 (SD: $15,283) in 2016, and median payments were from $1430 (IQR: $511–4531) in 2017 to $1737 (IQR: $642–$5286) in 2018. The payment values per specialist remained constant, with an average annual change of −1.2% (95% CI: −4.7% –2.3%, *p* = 0.49). The prevalence of specialists with payments decreased by −1.3% (95% CI: −2.9–0.4, *p* = 0.13) in each year from 47.1% (760 out of 1614) in 2016 to 44.9% (724 out of 1614) in 2019, but were not statistically significant (*p* = 0.12).

Among 78 companies making payments, 10 companies were devoid of the four-year continuous payment data. Excluding payments from ten companies, the specialists received payments averaging from $5562 (SD: $13,383) in 2017 to $6105 (SD: $13,312) in 2018. There were also no statistically significant annual changes in payments per specialist (average annual change rate: −1.3% [95% CI: −4.7–2.3%], *p* = 0.48) and prevalence of specialists with payments (average annual change rate: −1.4% [95% CI: −3.1–0.2%], *p* = 0.093) between 2016 and 2019 (Table 2).

### 4.3. Payment by Pharmaceutical Companies

The top companies made 63.8% of the total payments, representing $11,340,870 and 13,247 contracts (Figure 1). In company level analysis, the average and median number of specialists with payments per company were 74.9 (SD: 98.8) and 27.0 (IQR: 5.0–113.0), respectively. The average payments and number of contracts per specialist were $2333 (SD: $2578) and 2.8 (SD: 1.9) contracts, respectively. In short, each company made an average of $2333 payments, entailing 2.8 contracts per specialist, to 74.9 specialists in average for the reimbursement of lecturing, consulting and writing.

MSD made the largest payments of $2,493,244 to 460 (28.5%) specialists. Pfizer with the second largest payments distributed a total of $1,376,045 payments to 267 (16.5%) infectious specialists. The average payments per specialist were the highest from FujiFilm Toyama Chemical ($7269), followed by MSD ($5456), Pfizer ($5154), Boehringer Ingelheim ($5002), and AstraZeneca ($4990). Payment categories by each company were described in Appendix A. MSD also had the largest number of drugs with new and additional indications (8 drugs), followed by Daiichi Sankyo Company (5 drugs) and GlaxoSmithKline (5 drugs) (Appendix A). There were moderately positive correlations between number of new or additional indications and (1) total payments (r(76) = 0.46, *p* < 0.001) and (2) number of specialists with payments (r(76) = 0.43, *p* < 0.001).

### 4.4. Geographical Payment Distribution

There were geographical differences in distribution of infectious disease specialists (Appendix A). Number of infectious disease specialists per million populations ranged from 0.8 in Iwate Prefecture to 47.9 in Nagasaki Prefecture, while the average number of specialists per million was 12.7 in nationwide. There were geographic differences in total and per-specialist payment distribution as well (Appendix A). The average payment values per specialist were the highest in Okayama Prefecture ($21,750) and lowest in Ibaraki Prefecture ($1574).

In the analysis by region, the number of specialists per million populations ranged from 7.8 in the Hokkaido region (northernmost of Japan), and 8.7 in the Tohoku region (northernmost of main Japanese islands) to 20.6 in the Kyusyu region (southernmost of Japan). Meanwhile, the average payments per specialist were the highest in Tohoku region ($15,057), followed by Chugoku (the western part of main Japanese islands, $13,980) and Kyusyu regions ($13,394).

## 5. Discussion

This study demonstrated that a total $17,717,264 personal payments, equal to 1.8% of all payments were distributed to the board-certified infectious disease specialists over the period of four years in Japan. Among all Japanese board-certified infectious disease specialists, 65.4% (1055 out of 1614) of the specialists received a median of $3183 personal payments from 78 pharmaceutical companies between 2016 and 2019. The payments per specialist and proportion of specialists with at least one payment remained stable between 2016 and 2019.

First, this study found that there were substantial financial relationships among the board-certified infectious disease specialists and pharmaceutical companies in Japan. Although the prevalence of specialists with payments were similar to the previous findings, ranging from 62.0% among hematologists to 70.6% among medical oncologists [4,5,16,23], payment values per physician among the infectious disease specialists ($3183 in median four-year combined payments and $1430–$1720 in median single-year payments) were higher than all of the available evidence among pediatric oncologists ($2,961 in average) [23], pulmonologists ($2210 in median) [4], hematologists ($2471 in median) [23], and medical oncologists ($1103 in one-year median) [16] in Japan. Overall, compared to the previous studies, Japanese infectious disease specialists have relatively higher financial relationships with pharmaceutical companies. Considering the average annual salary among the Japanese internists was about forty million JYP in 2022 (equal to about 130,000 USD), the payments from pharmaceutical companies were small for the specialists’ salary. However, we could not understate the influence of small payments, as DeJong et al. revealed that only small amounts of payments, namely meals and beverages with mean values of less than $20, increased physicians’ prescriptions of brand-name drugs [24].

Second, we found that the payment values and prevalence of specialists with payments did not significantly change between 2016 and 2019. Kusumi et al. found that the pharmaceutical companies increasingly prioritized the payments to hematologists in Japan, with a 11.2% annual increase in payments per specialist [5]. Also, similar trends were observed by Murayama et al. among Japanese pulmonologists, with 7.8% annual increase in payments [4]. Our finding was different from these studies, indicating that the financial relationships among infectious disease specialists and pharmaceutical companies did not decline nor increase, but remained stable for the last four years. Although we found that there were many drugs newly approved or gained additional indications for infectious diseases, the Japanese government now recommends physicians to refrain from using new antibiotics to prevent antimicrobial-resistant bacteria. This trend in payments might be due to the demand for fewer use of new antibiotics.

Furthermore, we found that vast majority of payments disproportionately concentrated only on a small portion of the infectious disease specialists in Japan. Surprisingly, a small portion of the specialists included authoritative specialists such as leaders of the JAID and other medical societies. For example, the specialist with the largest payments ($711,965) was in various authoritative positions such as a full professor at a private medical university and a very influential television commentator for infectious disease [13]. Also, he was the current executive member of the JAID and other medical societies.

The specialist with the second-largest payments ($421,678) was also in authoritative positions such as a full professor at a national university and an executive or council member at several medical societies, including the JAID, and the Japanese Respiratory Society. He also served on public authorities as an author of the clinical practice guidelines for COVID-19 issued by the Japanese Ministry of Health, Labor and Welfare [7] and as a member of government scientific advisory committee.

The specialist with the fourth largest payments ($318,565) was the former president of the JAID who served from 2017 to 2020. He was also a current JAID executive member and the deputy chairperson of the Japanese government COVID-19 scientific advisory board, but his COI was not publicly disclosed by neither the JAID nor the Japanese government [12].

The receipt of substantial personal payments by executive members of medical societies was widely prevalent in Japan and other countries such as the US. Saito et al. found that 86.9% of Japanese executive members received a median of $7486 personal payments in 2016, and especially members specialized in internal medicine had higher financial relationships with pharmaceutical companies [25]. Moynihan et al. elucidated that 72% of the US influential medical society leaders had financial ties with pharmaceutical companies, [26] and that 93% of the leaders of Infectious Diseases Society of America received $31,805 in median total payments for six years, where the payments were the most prevalent of ten influential medical societies in the US [26]. Although we did not evaluate financial relationships during the tenure of the board membership, our findings indicated that the current board members of Japanese Association for Infectious Diseases had much larger financial relationships with pharmaceutical companies over the past several years, with 3.2 times higher median annual payments than those among board members of other Japanese medical societies or at least 4.5 times higher median annual payment values than that of the Infectious Diseases Society of America.

A number of studies have shown that financial relationships with pharmaceutical companies influence physicians’ behavior in prescribing drugs, [24,27,28,29,30] recommendations of clinical practice guidelines, [6,10,31,32,33] and comments on drugs in pharmaceutical advisory committees [34,35,36,37]. Pharmaceutical companies sometimes spend more payments for marketing less effective and less advantageous drugs [38,39] but with more harms to patients [40,41]. Despite these influences, the trends of the physicians’ acceptance of personal payments from industries are still common [18,42,43] and are even increasing in several specialties [4,5,23,44]. Improved transparency is required to reduce the undue influences of financial relationships with pharmaceutical companies on physician behaviors and potentially patients care, [28] to increase trust in healthcare, and to provide patients with more information about their treatment [45,46,47,48]. One recent systematic review reveals that financial COIs among the clinical practice guideline authors and society board members associated with favorable recommendations and opinion statements for the industry [31]. However, there is no consensus on how to manage the financial relationships, and how to increase independency of healthcare professionals toward their primary interest of treating patients based on their best knowledge and conscience [49]. Restriction of these personal payments to the specialists to a certain degree during the society board membership would be a simple and reasonable solution, but it is equally hard to implement by the professional medical societies when many of the society board members and societies themselves are financially tied to pharmaceutical companies. In the case of the JAID, financial COIs self-declared by the board members were not publicly disclosed, and there was no restriction of the financial relationships among pharmaceutical companies and the board-certified specialists as of now. At least, more transparency in financial relationships between the leading infectious diseases physicians and industry should be guaranteed to the public and the patients. Full disclosure of all payments from industry should be done by the society board members, as in the American College of Rheumatology and the American Society of Clinical Oncology. To assess the accuracy of disclosure by the board members, payment database published by independent journalism group and research institute will be helpful [50], as the American Society of Clinical Oncology and American Gastroenterological Association have already done for their disclosure of guideline authors [51,52].

This analysis has a few limitations. As we previously noted, our manual collection of payment data from 92 pharmaceutical companies’ webpages might have included unavoidable human errors, despite our careful cross-checks to exclude duplicate physicians from the data. Second, currently, pharmaceutical companies do not disclose their payments concerning meals, beverages, accommodations, travel and stock ownerships, according to the JPMA guidance. This could have underestimated the extent and prevalence of overall financial relationships among the specialists and industries. Third, the data disclosed by the JAID and pharmaceutical companies did not provide us many of detailed demographics of the specialists such as the specialists’ gender, affiliation characteristics, positions within their affiliations, and their academic and clinical performances. Also, the JAID did not publish the name list of infectious disease specialists for previous years. Therefore, there would have been influence of many unavoidable confounders on the personal payments at individual specialist level. However, our robust statistical analysis with GEE modeling has helped nullify effects of such confounders to some extent. Still, further studies should have elucidated the relations among the specialists’ characteristics and the personal payments. Finally, this study was based on the open-access payment data and Japanese board-certified infectious disease specialists. Thus, the payment magnitude and trend may not be exactly replicable to other countries’ specialists. However, this might serve as a pathway for prospective researchers to explore the same in other countries as well.

## 6. Conclusions

The majority of the certified infectious disease specialists received substantial personal payments for the reimbursements of lecturing, consulting and writing from the pharmaceutical companies in Japan. These financial relationships with those companies remained stable for the past four years in Japan. Furthermore, high ranked specialists such as those in the executive board had stronger financial ties with the companies. Further research is needed to evaluate professional activities affected by these financial relationships, and efforts to reduce the impact should be explored.

## Figures and Tables

**Figure 1 ijerph-19-07417-f001:**
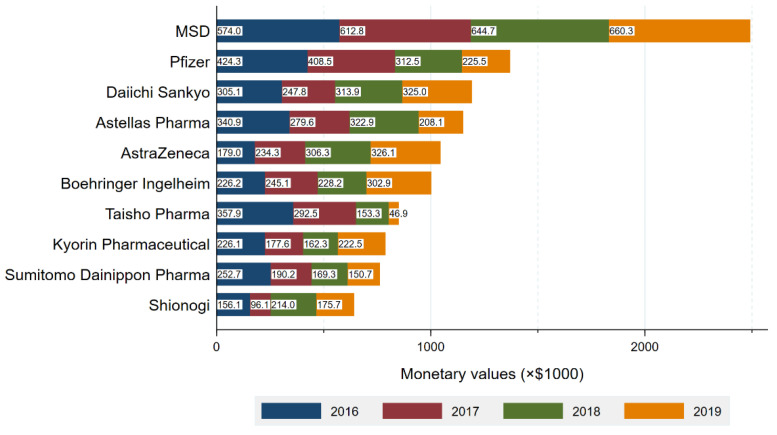
Total payment by company.

**Table 1 ijerph-19-07417-t001:** Summary of personal payments from Japanese pharmaceutical companies to infectious disease specialists certified by the Japanese Association for Infectious Disease between 2016 and 2019.

Variables	Number
Total	
Payment values, $	17,784,070
Contracts, n	21,680
Companies, n	78
Average per specialist (SD)	
Payment values, $	16,857 (45,010)
Contracts, n	20.5 (41.6)
Companies, n	5.6 (5.2)
Median per specialist (IQR)	
Payment values, $	3183 (938–11,250)
Contracts, n	6.0 (2.0–19.0)
Companies, n	4.0 (2.0–8.0)
Range	
Payment values, $	31–711,965
Contracts, n	1.0–538.0
Companies, n	1.0–29.0
Physicians with specific payments, n (%)	
Any payments	1055 (65.4)
Payments > $500	930 (57.6)
Payments > $1000	776 (48.1)
Payments > $5000	419 (26.0)
Payments > $10,000	290 (18.0)
Payments > $50,000	82 (5.1)
Payments > $100,000	43 (2.7)
Gini index	0.857
Category of payments	
Lecturing	
Payment value, $ (%)	14,607,478 (82.1)
Contracts, n (%)	18,078 (83.1)
Consulting	
Payment value, $ (%)	1,981,003 (11.1)
Contracts, n (%)	2122 (9.8)
Writing	
Payment value, $ (%)	797,929 (4.5)
Contracts, n (%)	1086 (5.0)
Other	
Payment value, $ (%)	397,659 (2.2)
Contracts, n (%)	459 (2.1)

**Table 2 ijerph-19-07417-t002:** Trends in personal payments from Japanese pharmaceutical companies to infectious disease specialists certified by the Japanese Association for Infectious Disease between 2016 and 2019.

Variables	2016	2017	2018	2019	Average Yearly Change (95%CI), %	*p*-Value	Combined Total
All pharmaceutical companies							
Total payments, $	4,662,217	4,215,566	4,538,520	4,367,767	–	–	17,78,4070
Average payments (SD), $	6134 (15,283)	5775 (13,410)	6108 (13,324)	6033 (11,837)	−1.2 (−4.7–2.3)	0.49	16,857 (45,010)
Median payments (IQR), $	1604 (511–4646)	1430 (511–4531)	1737 (642–5286)	1554 (662–5258)			3183 (938–11,250)
Payment range, $	92–216,035	92–160,610	95–190,726	31–144,593	–	–	31–711,965
Physicians with specific payments, n (%)							
Any payments	760 (47.1%)	730 (45.2%)	743 (46.0%)	724 (44.9%)	−1.3 (−2.9–0.4)	0.13	1055 (65.4)
Payments > $500	612 (37.9%)	594 (36.8%)	628 (38.9%)	616 (38.2%)	0.8 (−1.1–2.6)	0.43	930 (57.6)
Payments > $1000	482 (29.9%)	436 (27.0%)	485 (30.0%)	453 (28.1%)	−0.8 (−2.9%–1.3)	0.45	776 (48.1)
Payments > $5000	175 (10.8%)	178 (11.0%)	193(12.0%)	187 (11.6%)	2.8 (−0.8–6.5)	0.13	419 (26.0)
Payments > $10,000	106 (6.6%)	94 (5.8%)	113 (7.0%)	103 (6.4%)	1.0 (−3.6–5.7)	0.68	290 (18.0)
Payments > $50,000	16 (1.0%)	14 (0.9%)	17 (1.1%)	14 (0.9%)	−1.9 (−15.9–14.3)	0.80	82 (5.1)
Payments > $100,000	4 (0.2%)	3 (0.2%)	1 (0.1%)	2 (0.1%)	−28.3 (−52.4–8.1)	0.11	43 (2.7)
Gini index	0.878	0.881	0.870	0.876	–	–	0.860
Pharmaceutical companies with four-year payment data							
Total payments, $	4,597,653	4,205,920	4,492,988	4,314,421	–	–	17,610,982
Average payments (SD), $	6074 (15,169)	5562 (13,383)	6105 (13,312)	5992 (12,825)	−1.3 (−4.7–2.3)	0.48	16,788 (44,820)
Median payments (IQR), $	1603 (511–4642)	1430 (511–4525)	1737 (613–5280)	1552 (662–5258)			3183 (920–11,238)
Payment range, $	92–215,089	92–160,610	92–190,726	31–143,571	–	–	31–709,997
Physicians with specific payments, n (%)							
Any payments	757 (46.9)	730 (45.3)	736 (45.6)	720 (44.6)	−1.4 (−3.1–0.2)	0.093	1049 (65.0)
Payments >$500	609 (37.7)	593 (36.7)	620 (38.4)	610 (37.8)	0.5 (−1.4–2.4)	0.60	927 (57.4)
Payments > $1000	478 (29.6)	436 (27.0)	481 (29.8)	452 (28.0)	−0.7 (−2.8–1.4)	0.51	768 (47.6)
Payments > $5000	172 (10.7)	179 (11.1)	191 (11.8)	186 (11.5)	3.1 (−0.6–6.8)	0.10	415 (25.7)
Payments > $10,000	106 (6.6)	94 (5.8)	112 (7.1)	103 (6.4)	0.9 (−3.7–5.6)	0.71	289 (17.9)
Payments > $50,000	14 (0.9)	14 (0.9)	17 (1.1)	14 (0.9)	0.0 (−13.7–15.9)	1.0	82 (5.1)
Payments > $100,000	4 (0.2)	3 (0.2)	1 (0.1)	2 (0.1)	−28.3 (−52.4–8.1)	0.11	42 (2.6)
Gini index	0.879	0.881	0.871	0.876	–	–	0.860

IQR: interquartile range; SD: standard deviation.

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
