# Peer review of "Pharmaceutical Payments to Japanese Board-Certified Infectious Disease Specialists: A Four-Year Retrospective Analysis of Payments from 92 Pharmaceutical Companies between 2016 and 2019"

_ijerph, 2022, doi:10.3390/ijerph19127417_

Round 1
Reviewer 1 Report
The authors analysed the financial relationships between pharmaceutical companies and all infectious disease specialists board-certified by the Japanese Association for Infectious Disease. The findings are interesting. The reviewer has the following comments:
1) time scale: It is unclear to the reviewer why the authors decided to focus on the period between 2016 and 2019 only. Should the study period be extended to 10 years to obtain more widespread coverage?
2) Lagging: drug list between 2015 and 2019 was obtained while payment data were from 2016 to 2019. Has the lagging/ additional time from the new drug being approved to specialists receiving product-relevant payment been taken into account?
3) Discussion and Conclusion: The authors concluded that "such personal payments must be restricted to a certain level to avoid potential conflict of interest." The authors should detail how they extrapolate such conclusions from their data analysis. Also, mitigants or measures should be proposed in the Discussion section. The authors could refer to what measures have been taken from other jurisdictions in the world.
Author Response
Response to Reviewer 1
Comment
The authors analyzed the financial relationships between pharmaceutical companies and all infectious disease specialists board-certified by the Japanese Association for Infectious Disease. The findings are interesting. The reviewer has the following comments:
1) time scale: It is unclear to the reviewer why the authors decided to focus on the period between 2016 and 2019 only. Should the study period be extended to 10 years to obtain more widespread coverage?
Reply
We appreciate your interest in our study. As the companies disclosed and updated their payments on their company webpage each year, the payment data for previous years were not publicly disclosed after the disclosure period. Our research team has been collecting payment data from the companies continuously since 2019, when the 2016 payment data was disclosed. At the time of this study conducted, the latest analyzable, available payment data was payments for 2019. Therefore, we included all the analyzable and available payment data between 2016 and 2019 for this study. According to your comment, we have revised the manuscript, as follows.
Page 3, line 111-116
“The companies have published, updated the payment data each year on their company webpages, and deleted the payment data for previous years when they updated latest data. The payment data for all companies belonging to the JPMA were collected from the 2016 data. As of May 2022, the payment data in 2019 were the latest analyzable data in Japan. Thus, we could only analyze personal payments concerning lecturing, writing, and consulting between 2016 and 2019.”
2) Lagging: drug list between 2015 and 2019 was obtained while payment data were from 2016 to 2019. Has the lagging/additional time from the new drug being approved to specialists receiving product-relevant payment been taken into account?
Reply
Thank you for this comment. Considering the time lag between approval timing or the start on sale and the start of marketing to physicians, we considered that a predominant part of promotional activities for products approved in 2015 had started in 2016 and thereby included all new drugs approved for infectious diseases approved since 2015. We have added this explanation in the revised manuscript, as follows.
Page 3, line 99-105
“In order to take into account the time lag between approval timing and marketing timing by pharmaceutical companies, we considered that a predominant part of pro-motional activities for products approved in 2015 had started in 2016 and all drugs with additional or new indications for infectious diseases between 2015 and 2019 were extracted from the database of the Pharmaceuticals and Medical Devices Agency[15], the Japanese regulatory authority for drugs and medical devices.”
3) Discussion and Conclusion: The authors concluded that "such personal payments must be restricted to a certain level to avoid potential conflict of interest." The authors should detail how they extrapolate such conclusions from their data analysis. Also, mitigants or measures should be proposed in the Discussion section. The authors could refer to what measures have been taken from other jurisdictions in the world.
Reply
Thank you for this valuable comment on our study. We have added the recent systematic review assessing the association between recommendations in clinical practice guidelines and position statements endorsed by professional medical society and financial conflicts of interest of the physicians in order to support our conclusion, and also have revised conclusion, as follows.
Page 1, line 34-36
“There were substantial financial relationships between pharmaceutical companies and board-certified infectious disease specialists in Japan. Furthermore, high ranked specialists such as those in the executive board had stronger financial ties with the companies.”
Page 9, line 332-334
“One recent systematic review reveals that financial conflicts of interest among the clinical guideline authors and society board members associated with favorable recommendations and opinion statements for the industry.[31]”
Page 9, line 341-352
“In the case of the JAID, financial COIs self-declared by the board members were not publicly disclosed, and there was no restriction of the financial relationships among pharmaceutical companies and the board-certified specialists as of now. At least, more transparency in financial relationships between the leading infectious diseases physicians and industry should be guaranteed to the public and the patients. Full disclosure of all payments from industry should be done by the society board members, as in the American College of Rheumatology and the American Society of Clinical Oncology. To assess the accuracy of disclosure by the board members, payment database published by independent journalism group and research institute will be helpful,[50] as the American Society of Clinical Oncology and American Gastroenterological Association have already done for their disclosure of guideline authors.[51,52]”
Page 10, line 372-379
“The majority of the certified infectious disease specialists received substantial personal payments for the reimbursements of lecturing, consulting and writing from the pharmaceutical companies in Japan. These financial relationships with those com-panies remained stable for the past four years in Japan. Furthermore, high ranked specialists such as those in the executive board had stronger financial ties with the companies. Further research is needed to evaluate professional activities affected by these financial relationships, and efforts to reduce the impact should be explored.”

Reviewer 2 Report
Thank you for your effort and thorough analysis of the relations between infectious diseases specialists and the pharmaceutical industry in Japan. It is a consecutive article on payments received by Japanese physicians from the pharmaceutical sector in your portfolio. You have already studied several specializations and presented relevant results and conclusions in your previous research. The study does not add anything novel to already published papers in this context.
You do not name the phenomenon of accepting values by doctors for their favor or--if you wish--goodwill in the moment of prescribing or setting clinical practice guidelines.
It has already been named corruption, and numerous countries forbid such behavior as illegal or at least inappropriate. You consistently call it "payments" and suggest restricting it "to a certain level to avoid potential conflict of interest." This conclusion may be staggering for a European physician as any "payment" is regarded as unethical when there is a direct or indirect association with prescribing behavior, which you do not mention. You call a conflict of interest with pharmaceutical companies "contentious," a startlingly mild notion.
There is also the contentious issue of your self-citation behavior. Out of 50 references, 15 are by yourself. After publishing this paper, you will likely produce the next several for consecutive specializations. It makes a scientific buzz and is called an exploitation of the topic. The only difference would be the first author.
Despite your valuable effort, I will not recommend the journal promote your approach. There are significant methodological issues to be addressed. Instead of simplistic geographical analysis and multiplying publications, you should consider a multifactorial regression. That would present much more groundbreaking results without unjustified topic exploitation. But it would give only 2-3 articles instead of 16.
I have uploaded the manuscript with some additional comments.
Refering to Supplemental Material 1. Distribution of payment values per specialist
Please, change the X-axis range. The figures are illegible.
Refering to Supplemental Material 2. Payment concentration
Changing the order and starting the cumulative graph with the highest numbers would be more reasonable. Please, add N.

Author Response
Response to Reviewer 2
Comment
Thank you for your effort and thorough analysis of the relations between infectious diseases specialists and the pharmaceutical industry in Japan. It is a consecutive article on payments received by Japanese physicians from the pharmaceutical sector in your portfolio. You have already studied several specializations and presented relevant results and conclusions in your previous research. The study does not add anything novel to already published papers in this context.
Reply
Thank you for your review of our study. We partially agreed with your opinions, as our findings are consistent in some respects with our previous studies in other specialties and with other studies conducted in the United States and other industrialized countries. However, we still believe that the current findings are crucial given that a presence of infectious disease specialists has been enhanced in the global COVID-19 pandemic. We hope that this response letter will be of benefit to you and other readers of this journal in understanding our research.
You do not name the phenomenon of accepting values by doctors for their favor or--if you wish--goodwill in the moment of prescribing or setting clinical practice guidelines. It has already been named corruption, and numerous countries forbid such behavior as illegal or at least inappropriate. You consistently call it "payments" and suggest restricting it "to a certain level to avoid potential conflict of interest." This conclusion may be staggering for a European physician as any "payment" is regarded as unethical when there is a direct or indirect association with prescribing behavior, which you do not mention. You call a conflict of interest with pharmaceutical companies "contentious," a startlingly mild notion.
Reply
Thank you for this comment. We would like to note that payments disclosed by the companies, which we analyzed this time, were not made to directly urge the prescribing of specific drugs. They are mainly donations made to promote research and academic activities or payments for specific labors such as speaking and consulting activities. In this sense, while gratuities paid for the purpose of obtaining favors for one’s own companies can be considered bribes in Japan, the payments this time do not fit within such scopes. Nonetheless, a number of prior studies have shown that these payments are associated with unconscious prescribing patterns by physicians. However, as your comments, our study findings did not sufficiently support our conclusion, and we have revised the manuscript and conclusion, as follows.
Page 1, line 20-21
“Conflict of interest with pharmaceutical companies is one of the most concerned issues in infectious diseases.”
Page 1, line 34-36
“There were substantial financial relationships between pharmaceutical companies and board-certified infectious disease specialists in Japan. Furthermore, high ranked specialists such as those in the executive board had stronger financial ties with the companies.”
Page 10, line 372-379
“The majority of the certified infectious disease specialists received substantial personal payments for the reimbursements of lecturing, consulting and writing from the pharmaceutical companies in Japan. These financial relationships with those com-panies remained stable for the past four years in Japan. Furthermore, high ranked specialists such as those in the executive board had stronger financial ties with the companies. Further research is needed to evaluate professional activities affected by these financial relationships, and efforts to reduce the impact should be explored.”
There is also the contentious issue of your self-citation behavior. Out of 50 references, 15 are by yourself. After publishing this paper, you will likely produce the next several for consecutive specializations. It makes a scientific buzz and is called an exploitation of the topic. The only difference would be the first author. Despite your valuable effort, I will not recommend the journal promote your approach. There are significant methodological issues to be addressed.
Reply
As our research team is only sole team assessing the financial conflicts of interest among Japanese healthcare professionals using the payment data disclosed by the companies in Japan, there was a lack of documents concerning this issue in Japan from other researchers other than us. Therefore, we have cited several articles which we previously presented, in order to accurately explain the study context and rationale of this study in Japan. In the revised manuscript, we have reduced references from our previous studies and noted the rationale of our self-citation in the Conflicts of interest disclosure section, as follows.
Page 10, line 411-413
“Some of references in this study were self-cited by the study authors in order to explain the study context and rationale as well as to compare the findings with other studies. The other authors have no example of conflicts of interest to disclose.”
Instead of simplistic geographical analysis and multiplying publications, you should consider a multifactorial regression. That would present much more groundbreaking results without unjustified topic exploitation. But it would give only 2-3 articles instead of 16.
Reply
We appreciate your suggestion. In this study, we focused on the financial relationships between infectious diseases physicians and pharmaceutical companies, using the payment data made to the physicians publicly disclosed by major pharmaceutical companies in Japan. Furthermore, unlike the United States, there was few available demographic data from the pharmaceutical companies and JAID for the infectious disease physicians other than physicians’ practicing prefectures. Considering this limitation in the limited demographic data, we were forced to select a descriptive analysis of payments by prefecture rather than a multivariate regression analysis.
I have uploaded the manuscript with some additional comments.
Reply
Thanks for your comments. We have amended the manuscript accordingly.
Referring to Supplemental Material 1. Distribution of payment values per specialist. Please, change the X-axis range. The figures are illegible.
Reply
As the payment values were highly skewed, we have revised the Supplemental Material 1 with logarithmic X axis.
Referring to Supplemental Material 2. Payment concentration. Changing the order and starting the cumulative graph with the highest numbers would be more reasonable. Please, add N.
Reply
Thank you for this comment, but the Supplemental Material 2 is a Lorenz curve, which generally staring cumulative graph with lowest numbers and indicates concentration of a variable. Please refer to the article by Joseph L. Gastwirth. (https://www.jstor.org/stable/1937992?seq=2) We have added number of physicians in the graph.

Reviewer 3 Report
The manuscript consists of total 14 pages, including 2 tables, 1 figure and the list of total 50 references. The article is original material based study of financial flows from pharmaceutical companies to physicians specialized in infectious diseases in Japan. As such the article fits into the scope of works published in the Journal. It is an interesting topic in the context of the debate concerning the potential influence of pharmaceutical industry on physicians prescribing and treatment standards. The title is consistent with the contents of the main text of the manuscript.
The Introduction explains the background of the studied topic.
The Material and methods are explained in high detail.
The Results are consistent with the methodology accepted by the Authors and clearly presented.
However, it would be also interesting and most probably not too complicated, to present the Readers with some comparison between the payments made by the pharmaceutical industry to the physicians and the highest, lowest and average yearly incomes of the specialized physicians in Japan, preferably belonging in the same group that received various levels of financial benefits from the industry. It would make it possible to find out of how big real significance the fact of receiving the given sum of money can be for the given physician.
Assuming that the average yearly income of a specialized physician in Japan is more-less as high as that reported in statistics available in the Internet (the first in the row Google source providing exact numbers and some interpretations e.c.
https://cyclinghikes.com/how-much-do-doctors-make-in-japan/- average yearly income among physicians, including those not specialized in any discipline, of 16,000,000 YPY or 122,000 USD, and the highest earned by the top specialized professionals, scientists and functional figures most probably much bigger) it becomes much more challenging to suggest in the Discussion and the Conclusions that the amounts of pharmaceutical industry payments the physicians received according to the Authors' results, may have made as significant share of those Japanese physicians susceptible to trade off and seriously undermine their professional medical ethics pillars [therefore the sentences suggesting the unsupported insinuations shall be changed].
To the contrary to the Authors' statement in the Discussion, it is not surprising at all that the highest earnings for "lecturing, writing and consulting" are among the physicians who occupy the most "authoritative positions" (professors, commentators, associations leaders etc.) as it is a rather universal standard that prices rise steeply with the position on the ladder in every branch of science, business or politics [therefore the sentence shall be changed]. The statement in the Conclusion "personal payments must be restricted to a certain level to avoid potential conflict of interest" is not supported enough by the results of the Authors' study (therefore the sentence shall be changed).
The Authors study just demonstrated that a significant amount of money changed hands in a legal and registered way - in particular, the Authors did not provide evidence that guidelines or any other decisions, respectively set or made by the infectious diseases specialized physicians who were studied as those who received money from the pharmaceutical industry in Japan, were in the merit wrong and skewed by them on purpose, supposedly in order to allow the industry make more money.
Moreover, "potential" is a loaded word that is often used to make unsupported by any evidence, in majority false, accusations sound seriously and justify otherwise unjustifiable actions; and conflicts of interests exist everywhere in the world and capping financial flows at some level does not solve this issue at all but may be seriously unjust. Therefore, the payments from pharmaceutical industry to physicians shall in the first place be relevant to the input or work the physicians provide in exchange for the payments to the industry (as the Authors state they have analyzed "personal payments concerning lecturing, writing, and consulting") - and not just formally capped at some arbitrarily set level; just earning a lot of money is not a sin by itself, also in medicine. Taking the above into account, in my opinion the Authors shall seriously consider reconstructing the abovementioned conclusion statement to be more accurate to their presented study findings.
The references are numerous, recent enough and relevant to the topic of the manuscript. The Authors may consider mentioning the following aspects in the introductory part of their article:
- corruption in medicine as a phenomenon, as in e.c. https://doi.org/10.3390/medicines8090054
- pathologies in the drug market, s in e.c.
https://doi.org/10.3390/medicines8070036
- mistakes and pathologies remedies in drugs prescribing and emerging remedies to them, as in e.c. https://doi.org/10.3390/ijerph18137043
https://doi.org/10.3390/jcm8030305
https://doi.org/10.3390/pharmacy9020106 https://doi.org/10.3390/jcm9010014
https://doi.org/10.3390/pharmacy7020060
Author Response
Response to Reviewer 3
Comment
The manuscript consists of total 14 pages, including 2 tables, 1 figure and the list of total 50 references. The article is an original material-based study of financial flows from pharmaceutical companies to physicians specialized in infectious diseases in Japan. As such the article fits into the scope of works published in the Journal. It is an interesting topic in the context of the debate concerning the potential influence of pharmaceutical industry on physicians prescribing and treatment standards. The title is consistent with the contents of the main text of the manuscript. The Introduction explains the background of the studied topic. The Material and methods are explained in high detail. The Results are consistent with the methodology accepted by the Authors and clearly presented.
Reply
Thank you for your interest and careful consideration of our research. We hope that our revision and response letter would be beneficial to understand our study.
However, it would be also interesting and most probably not too complicated, to present the Readers with some comparison between the payments made by the pharmaceutical industry to the physicians and the highest, lowest and average yearly incomes of the specialized physicians in Japan, preferably belonging in the same group that received various levels of financial benefits from the industry. It would make it possible to find out of how big real significance the fact of receiving the given sum of money can be for the given physician.
Reply
We have added information on the annual income of Japanese physicians in the revised manuscript, as follows.
Page 8, line 271-273
“Considering the average annual salary among the Japanese internists was about forty million JYP in 2022 (equal to about 130,000 USD), the payments from pharmaceutical companies were small for the specialists’ salary.”
Assuming that the average yearly income of a specialized physician in Japan is more-less as high as that reported in statistics available in the Internet (the first in the row Google source providing exact numbers and some interpretations e.c. https://cyclinghikes.com/how-much-do-doctors-make-in-japan/-), average yearly income among physicians, including those not specialized in any discipline, of 16,000,000 JPY or 122,000 USD, and the highest earned by the top specialized professionals, scientists and functional figures most probably much bigger) it becomes much more challenging to suggest in the Discussion and the Conclusions that the amounts of pharmaceutical industry payments the physicians received according to the Authors' results, may have made as significant share of those Japanese physicians susceptible to trade off and seriously undermine their professional medical ethics pillars [therefore the sentences suggesting the unsupported insinuations shall be changed].
Reply
Thank you for this comment. We disagreed with this comment. As a number of previous studies assessing the financial relationships between physicians and pharmaceutical companies suggests that only small payments lead to more frequent prescription of drugs which were made payments by the companies, increase healthcare costs, and prescribe more brand-name drugs over generic alternatives. For example as DeJong et al. revealed, only small amounts of payments, namely meals and beverages with mean values of less than $20, increased physicians’ prescriptions of brand-name drugs. Considering the results supported by this and a number of similar studies, the effect of payments with small values cannot be understated, regardless of the physician salary. We have revised the manuscript, as follows.
Page 8, line 274-276
“However, as we could not understate the influence of small payments, as DeJong et al. revealed that only small amounts of payments, namely meals and beverages with mean values of less than $20, increased physicians’ prescriptions of brand-name drugs.[24]”
To the contrary to the Authors' statement in the Discussion, it is not surprising at all that the highest earnings for "lecturing, writing and consulting" are among the physicians who occupy the most "authoritative positions" (professors, commentators, associations leaders etc.) as it is a rather universal standard that prices rise steeply with the position on the ladder in every branch of science, business or politics [therefore the sentence shall be changed]. The statement in the Conclusion "personal payments must be restricted to a certain level to avoid potential conflict of interest" is not supported enough by the results of the Authors' study (therefore the sentence shall be changed).
Reply
Thank you for your comment. It is a very important point. It is generally known that conflicts of interest with industries with large amounts of money tend to be concentrated in a limited number of physicians. The problem is that unlike the compensation that one should normally receive for general labor, the compensation that comes from these industries to medical personnel can be a financial relationship with ulterior motives. In other words, these can have a promotional aspect and sometimes can be conflicts of interest between physicians’ primary mission, namely treating patients with their own best knowledge and consciousness, and financial interest from industries. Accumulating evidence suggest having a financial relationship with industry, whether or not it is intended by the medical personnel, can influence the medical practice of them. On the other hand, as you pointed out, it is understandable to argue that remuneration always involves labor such as lecture, writing, and consulting, and therefore should be commensurate with it, and that physicians with authority should be paid more. However, considering that the authority figures discussed in this paper are the leaders of academic societies and that their activities influence the medical practices and guidelines of many physicians, it is the physicians with authority who need to consider the nature of their relationship with the industry. Based on your remarks, we have modified the wording of our conclusions as follows.
Page 1, line 34-36
“There were substantial financial relationships between pharmaceutical companies and board-certified infectious disease specialists in Japan. Furthermore, high ranked specialists such as those in the executive board had stronger financial ties with the companies.”
Page 10, line 374-377
“Furthermore, high ranked specialists such as those in the executive board had stronger financial ties with the companies. Further research is needed to evaluate professional activities affected by these financial relationships, and efforts to reduce the impact should be explored.”
The Authors study just demonstrated that a significant amount of money changed hands in a legal and registered way - in particular, the Authors did not provide evidence that guidelines or any other decisions, respectively set or made by the infectious diseases specialized physicians who were studied as those who received money from the pharmaceutical industry in Japan, were in the merit wrong and skewed by them on purpose, supposedly in order to allow the industry make more money.
Reply
Thank you for your comment. As you point out, this paper, by its design, is not intended to elucidate the causal relationship between financial relationships and decisions on guidelines, etc. In other words, this study is about the amount of payments from pharmaceutical companies to physicians, its prevalence, and trends. From the results of many previous studies, it is already a consensus that even small monetary relationships can influence medical practice, and an increasing number of countries and academic societies have established rules to ensure the neutrality of guidelines. Many studies have been conducted mainly in Europe and the U.S. We believe that this study is useful for international comparisons and for understanding the current situation in Japan.
Moreover, "potential" is a loaded word that is often used to make unsupported by any evidence, in majority false, accusations sound seriously and justify otherwise unjustifiable actions; and conflicts of interests exist everywhere in the world and capping financial flows at some level does not solve this issue at all but may be seriously unjust. Therefore, the payments from pharmaceutical industry to physicians shall in the first place be relevant to the input or work the physicians provide in exchange for the payments to the industry (as the Authors state they have analyzed "personal payments concerning lecturing, writing, and consulting") - and not just formally capped at some arbitrarily set level; just earning a lot of money is not a sin by itself, also in medicine. Taking the above into account, in my opinion the Authors shall seriously consider reconstructing the abovementioned conclusion statement to be more accurate to their presented study findings.
Reply
We have carefully read your comments. As you point out, conflicts of interest exist in many fields, not only in medicine. In particular, it is only in recent years that conflicts of interest have become an issue in medicine, and this is based on the lessons learned from research misconduct and the unfortunate patients that have resulted from it. However, we have to note that healthcare professionals should prioritize patients’ benefits more than anything else and interactions with the industry should be limited to something related to research and development. Indeed, we have experienced multiple scandals the roots of which exists in such financial relationships. In recent years, the opioid crisis is a good example. Setting a cap on the amount of compensation physicians receive from industry and academia is only one possible solution. What is important is to increase transparency and to further the discussion in the medical community and industry about appropriate relationships and management. As noted, the concluding sentences have been revised as follows.
Page 1, line 34-36
“There were substantial financial relationships between pharmaceutical companies and board-certified infectious disease specialists in Japan. Furthermore, high ranked specialists such as those in the executive board had stronger financial ties with the companies.”
Page 10, line 374-377
“Furthermore, high ranked specialists such as those in the executive board had stronger financial ties with the companies. Further research is needed to evaluate professional activities affected by these financial relationships, and efforts to reduce the impact should be explored.”
The references are numerous, recent enough and relevant to the topic of the manuscript. The Authors may consider mentioning the following aspects in the introductory part of their article: - corruption in medicine as a phenomenon, as in e.c. https://doi.org/10.3390/medicines8090054
- pathologies in the drug market, s in e.c.
https://doi.org/10.3390/medicines8070036
- mistakes and pathologies remedies in drugs prescribing and emerging remedies to them, as in e.c. https://doi.org/10.3390/ijerph18137043
https://doi.org/10.3390/jcm8030305
https://doi.org/10.3390/pharmacy9020106
https://doi.org/10.3390/jcm9010014
https://doi.org/10.3390/pharmacy7020060
Reply
Thank you very much. We have read the literature and have added the following reference to the introduction of the text.
Page 1, line 43-45
“There is concern that pharmaceutical companies are attempting to influence prescribing and practice activities through lobbying activities against physicians and academic activities.[1]”
Reference
Ethics in Medicines: Exposing Unethical Practices and Corruption in All Sectors of Medicines Is Essential for Improving Global Public Health and Saving Patients’ Lives. Medicines 2021,8(9) 54
Page 2, line 50-53
“While there have been efforts to uncover inappropriate prescribing and how to deal with it, efforts to properly manage financial conflicts of interest are ultimately the most important in the series of activities to date that encourage appropriate prescribing activities.[8,9]”
Reference
Nunez-Montenegro A, et al. Evaluation of Inappropriate Prescribing in Patients Older than 65 Years in Primary Health Care, J.Clin. Med 2019, 8(3),305
Diaz S et.al. Changes in Opioid Prescribing Behaviors among Family Physicians Who Participated in a Weekly Tele-Mentoring Program. Clin. Med. 2020,9(1),1
